# Use of *Lupinus albus* as a Local Protein Source in the Production of High-Quality Iberian Pig Products

**DOI:** 10.3390/ani14213084

**Published:** 2024-10-25

**Authors:** Javier García-Gudiño, Montaña López-Parra, Francisco Ignacio Hernández-García, Carmen Barraso, Mercedes Izquierdo, María José Lozano, Javier Matías

**Affiliations:** Centre of Scientific and Technological Research of Extremadura (CICYTEX), Junta de Extremadura, 06187 Guadajira, Spain; javier.garciag@juntaex.es (J.G.-G.); montana.lopez@juntaex.es (M.L.-P.); carmen.barraso@juntaex.es (C.B.); mercedes.izquierdo@juntaex.es (M.I.); mariajose.lozano@juntaex.es (M.J.L.); javier.matias@juntaex.es (J.M.)

**Keywords:** animal nutrition, sustainability, environmental impacts, consumers

## Abstract

In this study, we explored the potential of using sweet white lupin (*Lupinus albus* L.) seeds instead of soybean meal in the diets of Iberian pigs. The results showed that the lupin-based diet performed as well as the conventional soybean meal diet without affecting the pigs’ growth, carcass characteristics, or meat quality. In particular, pigs fed lupin had lower levels of saturated fatty acids in their meat, particularly in the pre-finishing phase, which could make the meat healthier. Overall, lupin offers a sustainable and beneficial alternative to soybean meal, improving both environmental impact and meat quality in Iberian pig production.

## 1. Introduction

Spain is the fourth largest pork producer in the world and the leading producer in Europe [1,2]. Although conventional pig production systems predominate in Spain, they coexist with the Iberian pig production system, which includes both extensive and intensive farming systems [3]. The Iberian breed is a local breed traditionally reared in the south-western region of the Iberian Peninsula [4]. It is mainly reared on natural resources within the *dehesa* ecosystem [5] and is particularly known for the high quality of its cured products [6].

The development of the Spanish economy and the globalisation of trade have contributed significantly to the growing demand for high-quality and traditionally produced Iberian cured products [7]. Nevertheless, the limited availability of *dehesa* surface limits the number of pigs that can be reared solely on natural resources [6]. For this reason, Iberian pig production has expanded geographically throughout the country, also adopting a conventional production system [8].

Iberian pork products have improved their market position in recent years, increasing in importance and acceptance in national and international markets [9]. While consumer preference remains strongly oriented towards traditional cured products from free-range acorn-fed Iberian pigs [10], it is essential to explore new market opportunities through the development of innovative Iberian pork products.

In recent years, there is an increasing demand for fresh meat from Iberian pigs [11,12], mainly due to its exceptional organoleptic characteristics [13,14,15]. To produce only fresh meat, the production cycle of Iberian pigs could be shortened by slaughtering the animals at the end of the growing phase. This type of production is referred to as *primor* pig production [16], in which slaughter takes place at a body weight of only 90–100 kg. In contrast, to obtain cured products, it is necessary to fatten the animals to a body weight of about 165 kg [17], which requires a long production cycle of about 10 to 14 months, depending on the management and genotype used during the fattening phase [18].

It is therefore essential to adapt Iberian pig production to new market demands, taking into account the strong environmental and ethical values that consumers associate with the high-quality Iberian pig products. In fact, consumer choices are influenced by extrinsic values such as local production [19], animal welfare [20], or sustainable food production [21]. Iberian pig products offer different added values, such as local production [22] and a high level of animal welfare [23], which are not commonly found in conventional white pig products. However, in terms of sustainable production, Iberian pig production has a higher environmental impact per kg of live pig at farm gate than other pig production systems [24,25], mainly due to its long production cycle [8], which includes the rearing period before fattening.

The most significant environmental impacts of Iberian pig production occur during the growing phase [25], mainly due to the higher protein requirements during this phase [26] and the dependence on imported soy [27]. Therefore, it is crucial to explore alternative local protein sources for Iberian pig production in order to improve sustainability. Among the different local alternatives, lupine is emerging as a promising option due to its high protein content [28]. Previous studies have shown that lupine can be an effective source of protein in livestock systems [29,30,31]. However, the use of lupine in Iberian pig production has not been previously promoted so far due to the presence of antinutritional factors [32]. Nevertheless, the development of new varieties with lower levels of antinutritional factors opens the door to renewed research into its inclusion in Iberian pig diets.

The general aim of this work was to evaluate the use of lupine as an alternative protein source in the production of high-quality Iberian pig products, with a focus on improving sustainability. The specific objectives were: (a) to evaluate growth and metabolic parameters during the pre-finishing and fattening phases in Iberian pigs under a concentrate diet based on soy or in lupine, and (b) to evaluate the carcass yield and the quality of the meat obtained during these different stages of production.

## 2. Materials and Methods

### 2.1. Agronomical Studies on Lupinus albus and Experimental Pig Diets

An area of one hectare was sown in November on the agricultural farm “La Orden” (CICYTEX; +38°51′2.5′′, −6°40′14.7′′) with the *Orden Dorado* variety of sweet white lupine (*Lupinus albus* L.). This variety, developed by CICYTEX [33] through classical genetic breeding, involved crossing a Polish sweet lupin variety (*Lupinus albus* L.) with a local bitter variety, with alkaloid content monitored by the Drangendorff method [34], was sown at a density of 120 kg/ha (approximately 30 seeds/m^2^). Before sowing, the soil was harrowed and fertilised with 16, 30, and 30 kg/ha N, P_2_O_5_, and K_2_O, respectively. A pre-emergence herbicide (pendimethalin 33%) was applied at 3.5 L/ha. Germination occurred in December and flowering in March. Harvesting took place in June, using a Wintersteiger harvester for the plots under study, with an average yield of 1500 kg/ha. The harvested grain had an average humidity, crude protein and crude fat values of 7.5%, 42.7%, and 6.5%, respectively. The harvested grain was properly stored until the elaboration of the concentrate feedstuff used in the present study.

Pre-finishing and fattening diets were formulated according to FEDNA’s specifications [35]. Grower pigs were fed a control diet containing 68.9 g/kg soybean meal or a diet in which soybean meal was replaced by an amount of 105 g/kg of lupine grain in order to achieve the same protein content in both growth diets. Similarly, soybean meal was replaced by lupine grain as protein source in the fattening diets (35 g/kg compared to 50 g/kg, respectively). The protein requirements were adapted to each of the production phases of the Iberian pig [35]. The nutritional and ingredient composition of the different diets are shown in Table 1.

### 2.2. Animals and Experimental Design

A total of 50 castrated male Iberian pigs were used in this study and reared under semiextensive conditions in the Valdesequera farm (+39°3′13.2′′, −6°50′45.2′′), belonging to CICYTEX. As shown in Figure 1, the experiment was carried out during the growing and fattening phase, from 6 to 14 months of age.

At the start of the experiment, animals were randomly assigned (stratified by body weight, BW) to one of two groups (n = 25 pigs), each having a similar mean BW (*p* > 0.1). The treatment groups were as follows: (1) Soya group (Control; SG; n = 25), fed a standard growth diet (containing 6.89% soya meal; Table 1) until the end of the growing phase. (2) Lupine group (Treated; LG; n = 25), fed the same type of diet (but with 10.5% *Lupinus albus* L.; Table 1) until the end of the growing phase. At the end of the growth phase, at 11 months of age, 10 animals (*primor* pigs) from each group were slaughtered to obtain fresh meat samples and other zootechnical parameters. Therefore, during the fattening phase, the treatment groups were reduced to 15 animals per group, with SG (n = 15) fed with a standard fattening diet (containing 3.5% soya meal; Table 1) and LG (n = 15) fed with the same type of diet (but containing 5% *Lupinus albus* L.; Table 1) until 14 months of age, when the fatteners were slaughtered to collect meat samples and other zootechnical parameters. Pigs were housed in large outdoor corrals throughout the experiment. The animal research protocols were reviewed and approved by the Ethics Committee of the University of Extremadura (Spain; reference 79/2023).

### 2.3. Growth Measurements, Carcass Traits and Sample Collection

Body weight was collected at the beginning of the experiment (approximately 6 months of age) and then monthly. Ultrasound scans were performed at the beginning and end of the growth phase and at the end of the fattening phase. During the ultrasound scans, the pigs were loosely restrained in a crate to minimise movement and maintain a standing posture. Ultrasound images were obtained using an EXAGO ultrasound scanner (Imv Technologies, L’Aigle, France) equipped with a 13 cm, 3.5 MHz linear probe. Longitudinal scans were taken at approximately 7 cm from the dorsal midline over at least the 10th and 14th ribs, and transversal scans were done over the intercostal space between the 10th and 11th ribs. The distance (loin depth) in the *Longissimus dorsi* muscle from the dorsal (upper) surface to the ventral (lower) surface, the thickness of subcutaneous fat (back fat) over the *Longissimus dorsi* muscle in the transversal image, and the transversal surface area (loin area) were measured at the level of the 10th rib [36]. Image analysis was performed using the software BioSoft Toolbox^®^ II for Swine (Biotronics Inc., Ames, IA, USA, https://www.humeco.net/producto/software-biosoft-toolbox-ii, accessed on 14 September 2024).

Blood samples were collected in EDTA-coated vacuum tubes by puncturing the orbital venous sinus at the beginning and at the end of the growing phase, and at the end of finishing, before slaughter. Blood samples were centrifuged at 1500× *g* for 10 min at 21 °C to separate the plasma. Biochemical analyses were conducted on the plasma samples at the University of León’s laboratory (Spain) using a BA400 automatic analyser (Biosystems, Barcelona, Spain) to measure the following parameters: alkaline phosphatase (ALP), alanine aminotransferase (ALT), aspartate aminotransferase (AST), gamma-glutamyl transferase (GGT), protein (total), bilirubin (direct and total), cholesterol, high-density lipoprotein (HDL), low-density lipoprotein (LDL), and triglycerides.

After the slaughter of *primor* and fattening pigs, at approximately 11 and 14 months of age, respectively, the weights of carcasses and prime cuts (hams, forelegs, and loins) were recorded. Specifically, carcass weight (cw) was measured after slaughter and dressing. Ham weight (hw) was determined by removing the hind legs at the hip joint, foreleg weight (fw) by separating the front legs at the shoulder joint and loin weight (lw) by removing the muscle along the spine between the shoulder and the hip. The prime cut weight (pcw) was calculated as the sum of the ham, foreleg and loin weights (pcw = hw + fw + lw). The yields of carcass (cy), hams (hy), forelegs (fy), loins (ly), and total prime cuts (pcy) were calculated using the following formulas:cy = (cw/BW) × 100
hy = (hw/cw) × 100
fy = (fw/cw) × 100
ly = (lw/cw) × 100
pcy = (pcw/cw) × 100

These calculations were performed as described in [37] to ensure a standardised methodology for evaluating carcass and cut yields. For the analyses of meat quality, the left loin was taken from each animal.

### 2.4. Meat Quality Analyses

For each loin sample, instrumental colour and pH value were measured. The colour parameters, including lightness (L*), redness (a*), and yellowness (b*) in the CIE Lab colour space, were determined using a hand-held colorimeter (Minolta CR-400, Konica Minolta, Osaka, Japan), according to the method described by Girolami et al. [38]. pH was measured using a portable puncture pH meter (model pH25+ portable, Crison Instruments, Barcelona, Spain).

In addition, intramuscular fat (IMF) was determined by the method of Folch et al. [39], in which meat samples were successively washed with a mixture of chloroform and methanol (2:1), the solvent was evaporated, and the extracted fat was weighed. In addition, the fatty acid profile of the total lipid extract was then determined using the method described by Morrison and Smith [40]. Analysis of fatty acid methyl esters (FAMEs) was conducted using an Agilent 6890 gas chromatograph (Agilent Technologies, Santa Clara, CA, USA) equipped with a flame ionization detector (FID) and a silica column DB-23 (60 mm length, 0.25 mm inner diameter, and 0.25 μm film thickness). The injector and detector temperatures were set at 260 °C and 280 °C, respectively. The oven temperature was programmed to reach 220 °C. Helium was used as the carrier gas at a constant flow rate of 1.2 mL min^−1^ with a make-up flow of 25 mL min^−1^. A split injection mode with a ratio of 1:100 was employed. Identification of individual FAMEs was performed by comparing retention times with those of standard FAME mixtures (Supelco 37 Component FAME Mix, Sigma Aldrich, St. Louis, MO, USA). The amount of each fatty acid was calculated on the total of fatty acids detected and expressed as g/100 g of FAMEs, and the different fatty acid groups were obtained: sum of all saturated fatty acids (SFA) detected (C12:0 + C14:0 + C16:0 + C17:0 + C18:0 + C20:0); sum of all monounsaturated fatty acids (MUFA) detected (C16:1 n-9 + C16:1 n-7 + C17:1 + C18:1 n-9 + C18:1 n-7 + C20:1 n-9 + C20:1 n-7); and sum of all polyunsaturated fatty acids (PUFA) detected (C18:2 n-6 + C18:3 n-4 + C18:3 n-3).

The texture analysis of the loin samples was only carried out on the *primor* pig samples, as these were the only samples intended for fresh meat consumption. Texture analysis was performed in a TA XT-2i texture analyser (Stable Micro Systems Ltd., Surrey, UK) according to Combes et al. [41]. For texture profile analysis (TPA), uniform portions of the loin were cut into 10 mm cubes. The following texture parameters were measured according to Bourne et al. [42]: hardness (N/cm^2^); springiness (cm); gumminess (N); chewiness (N cm s^2^); and cohesiveness (dimensionless).

### 2.5. Statistical Analyses

To evaluate the effects of the treatments on the animals, data were collected on pig growth, blood parameters, carcass traits, and meat quality. The statistical model included the fixed effect of dietary treatment, with each pig considered as an experimental unit. One-way analysis of variance (ANOVA) was performed using the general linear models (GLM). Analyses were performed using SAS software version 9.4. Results are presented as least squares means (LSMeans) for each treatment and root mean square error (RMSE), with a significance level of *p* < 0.05.

## 3. Results

### 3.1. Growth, Ultrasound Measurements and Biochemical Parameters

The BW of the Iberian pigs in the different phases of the experiment is shown in Table 2. The BW during growing and fattening phases was not affected by diet (*p* > 0.4). Average daily gain (ADG) did not differ between groups in the different production phases. However, due to the lack of replication in the experimental design, the results related to growth parameters should be interpreted with caution and considered as indicative rather than conclusive.

Table 3 shows the carcass traits predicted in vivo by ultrasounds during the growth and fattening phases: loin depth, back fat, and loin area at the level of the tenth thoracic rib. No differences were found between the different ultrasound measurements at the start of the growing phase (*p* > 0.1), at the end of this phase (*p* > 0.1), or at the end of the fattening phase (*p* > 0.3).

Table 4 shows biochemical parameters in blood plasma: alanine aminotransferase, bilirubin, cholesterol, and triglycerides. The levels of bilirubin and cholesterol did not differ between the groups during the two productive phases. However, triglyceride levels were higher in SG (*p* = 0.005) at the end of the growing phase and alanine aminotransferase levels were higher in SG (*p* = 0.001) than in LG at the end of the fattening phase. Total protein, total bilirubin, HDL, and LDL did not differ between groups.

### 3.2. Carcass Traits and Meat Quality in Iberian Primor (Prefinishing) Pigs

Table 5 shows the carcass traits of the Iberian *primor* pigs: carcass, hams, forelegs, and loin weights showed no differences between groups (*p* > 0.1). Similarly, there were no group differences in the yields of carcass, prime cuts, hams, forelegs, and loins (*p* > 0.1).

Table 6 shows the results of the meat quality analyses conducted on the *Longissimus dorsi* muscle of Iberian *primor* pigs: colour parameters (L*, a*, b*) and pH showed no differences between the groups (*p* > 0.2). No significant differences were observed for intramuscular fat (*p* = 0.058). However, in terms of fatty acid content, a higher SFA content was found in SG (*p* = 0.007). Both MUFA and PUFA showed no differences between groups (*p* = 0.099 and *p* = 0.394, respectively).

The texture parameters (springiness, gumminess, chewiness, and cohesiveness) did not differ between groups.

### 3.3. Carcass Traits and Meat Quality in Iberian Pigs After Finishing

Table 7 shows the post-mortem carcass traits of the Iberian fattening pigs: weights (carcass, ham, foreleg, and loin) and yields (carcass, prime cut, ham, foreleg, and loin) showed no differences between groups (*p* > 0.2 and *p* > 0.5, respectively).

Table 8 shows the results of the meat quality analyses conducted on the *Longissimus dorsi* muscle of Iberian fattening pigs: colour parameters (L*, a*, b*) and pH showed no differences between groups. No significant differences were observed between groups for intramuscular fat and fatty acid content (SFA, MUFA, and PUFA).

## 4. Discussion

In the meta-analysis carried out by [43] on protein sources alternative to soybean meal in pig diets, no significant differences were found in the production parameters when lupin was compared with a traditional diet based on soybean meal. Consistent with these findings, our study found that the replacement of soybean meal with sweet white lupin (*Lupinus albus* L.) seeds in the concentrate of Iberian pigs did not significantly affect performance in terms of body weight during either the growing or the fattening phases. This is in agreement with the studies by [44,45] in a cosmopolitan pig breed. Adequate protein ingestion during the growing phase is particularly important considering the high protein requirements during this period [35], especially as Iberian pigs tend to develop more lean tissue and less fat during the growing phase [46].

Similarly, the carcass traits predicted in vivo by ultrasound during the growing and fattening phases were not affected by the replacement of soybean meal by lupin in the diets. The ultrasound measurements obtained in this study were consistent with those reported in other studies during these phases [47,48]. However, our results showed a larger loin area and lower backfat thickness at the tenth thoracic rib level compared to the findings of Ayuso et al. [36,49] in Iberian pigs fattened on natural resources in the *dehesa* ecosystem. This difference is probably due to the lower protein content of acorns [50], which contributes to greater fat deposition and lower muscle accretion in free-range acorn feeding Iberian pigs compared to those reared under other feeding systems [49].

Sobotka et al. [51] reported no significant effects on carcass traits when yellow lupine was used as an alternative vegetal protein source to partially replace soybean meal in grower and finisher pigs, which is consistent with our results in Iberian pigs. In addition, the mean values for carcass traits obtained in the present study are consistent with previous reports on Iberian pigs [37,49]. While carcass yield increases significantly with slaughter weight, prime cut yields tend to decrease with increasing slaughter weight in Iberian pigs [37] due to higher fat deposition in heavy pigs [46]. This explains why the fattened Iberian pigs in our study had a higher carcass yield compared to *primor* pigs. Conversely, consumption of the *Orden Dorado* variety of white lupine (low in antinutritional factors) did not adversely affect liver function, as observed in the blood biochemical parameters obtained. Similarly, Sobotka et al. [51] did not observe any alteration in liver function with the consumption of yellow lupine, although in their study the inclusion of lupine in the diet was limited to 6%.

Overall, the productive results observed in this study, in line with previous research in other pig breeds [30,51], suggest that lupin (*Lupinus albus* L.) is a viable alternative to soybean meal in Iberian pig production from an environmental and economic point of view. Specifically, lupin production contributes to reducing emissions and improving resources efficiency [27], thus addressing key environmental concerns. Economically, lupin’s adaptability to different climates, poor soils, and adverse conditions [52] makes it a competitive crop, with the potential to generate comparable or even higher profits than other traditional crops in Spain [53], in addition to being a rainfed crop, contrarily to soybean, which usually needs irrigation in temperate and mediterranean climates.

Furthermore, in the present study the use of lupine in Iberian pig diets has shown additional benefits in terms of fresh meat quality, particularly in meat from *primor* pigs, i.e., those slaughtered at the end of the growing phase. Firstly, fresh meat from Iberian pigs had a higher intramuscular fat content in both *primor* and fattening pigs compared to other cosmopolitan pig breeds, including the Duroc breed [54,55]. Higher intramuscular fat content contributes to improved sensory attributes, in particular an increase in juiciness and tenderness of fresh meat [56]. Furthermore, the higher a* values obtained in Iberian fresh meat indicate a more intense red colour, which contrasts with the paler appearance typically observed in fresh meat from white pigs [54]. The a* values obtained in this study are in line with those reported in other studies on Iberian fresh meat [57,58]. Finally, significant differences in SFA content were observed in this meat, with lower levels of SFA in pigs fed lupine-based diets compared to those fed with soybean meal-based diets.

Spielmann et al. [59] demonstrated that lupin reduces plasma triglycerides in other monogastric species and further confirmed that protein isolated from *Lupinus albus*, the species used in our study, has a potent triglyceride-lowering effect. Our results are consistent with these findings, as we observed lower triglyceride levels at the end of the growing phase in pigs fed a lupin-based diet in comparison to those fed soy-based concentrate. Our results are also consistent with the work of Muñoz et al. [60], who reported a positive correlation between blood triglyceride levels and SFA content in meat. Given that SFA in meat has been associated with potential adverse health effects [61], the lower SFA content in the intramuscular fat of pigs fed lupin suggests a possible health benefit from consuming this meat. However, this reduction in SFA was not observed in meat from pigs reared to 165 kg BW, nor were there any differences in triglyceride levels between the study groups after finishing. This could be attributed to the lower inclusion rates of both lupin and soybean meal in finishing diets, a common practice due to reduced protein requirements in the fattening phase [35].

In addition to producing healthier fresh meat, rearing *primor* pigs has been identified as an efficient approach when only fresh pork production is intended form Iberian pigs. Extending the slaughter age reduces the average daily feed intake (ADFI) and subsequently the average daily gain (ADG) [62], leading to an overall increase in total feed consumption over the entire period, resulting in a decrease in the feed conversion ratio (FCR). By focusing on *primor* pigs, fresh Iberian pork can be obtained with lower feed consumption due to the shortened production cycle. This approach also results in a higher percentage of lean tissue at a lower slaughter weight [46], improving overall production efficiency and reducing environmental impacts. However, it is still necessary to fatten the pigs to produce Iberian cured products [8]. Although the inclusion of lupin in the diet of fattening pigs does not result in healthier meat, lupin remains an optimal feeding strategy for reducing the environmental impacts of the production of Iberian cured products.

In terms of consumer product choice, it has been shown that differentiated attributes in meat products significantly influence purchase decisions [63], which could boost sales of Iberian products. Attributes related to sustainability and health, such as those derived from lupin-based diets in this study, are increasingly prioritised by consumers [19,64]. Moreover, products derived from sustainable production systems tend to command higher prices, as consumers are willing to pay more for goods that are in line with their values [65]. Therefore, the use of lupin-based diets could improve not only the sustainability of Iberian pork production, but also consumer perception and preference for Iberian pork products, providing a competitive advantage in a market that increasingly values environmental responsibility and product quality.

Products derived from *primor* pigs offer a significant opportunity for the expansion of the Iberian market, both nationally and internationally. These products can open up new market niches, targeting consumers looking for food with healthier and more sustainable attributes. By positioning fresh Iberian meat from *primor* pigs as a healthier and more environmentally friendly alternative within the market, the consumer base can be broadened beyond the traditional focus on cured products. In addition, diversifying the Iberian product range not only enriches the options available to consumers, but also reduces the sector’s reliance on cured products such as ham, making the Iberian sector more stable and resilient to market fluctuations.

Given the promising results observed with the use of lupin as an alternative protein source in Iberian pig diets, future research could extend these findings to other livestock species. Investigating the potential benefits of lupin in the diets of ruminants, poultry, and other monogastric species could provide valuable insights into its wider applicability in animal nutrition. In addition, it would be important to assess the effect of lupin supplementation on the quality and nutritional profile of other animal products, such as milk and eggs. Understanding how lupin affects these products could enhance its utility in sustainable agriculture and diversify its use across livestock production systems.

## 5. Conclusions

The results of this study show that lupine (*Lupinus albus* L.) is a viable alternative to soybean meal as a protein source in the diet of Iberian pigs, offering several advantages without compromising production parameters. In particular, the lupin-based diet resulted in lower levels of saturated fatty acids (SFA) in the fresh meat of pre-finishing Iberian pigs (*primor pigs*), which could be beneficial for the production of healthier fresh meat products. Furthermore, in a market where consumers increasingly prefer healthier and sustainably produced foods, the inclusion of lupin in Iberian pig feed could provide this added value. Products derived from this practice would not only be in line with these consumer trends, but may also stand out in a market where quality and sustainability are key differentiators, thereby increasing the competitiveness of the Iberian sector.

## Figures and Tables

**Figure 1 animals-14-03084-f001:**
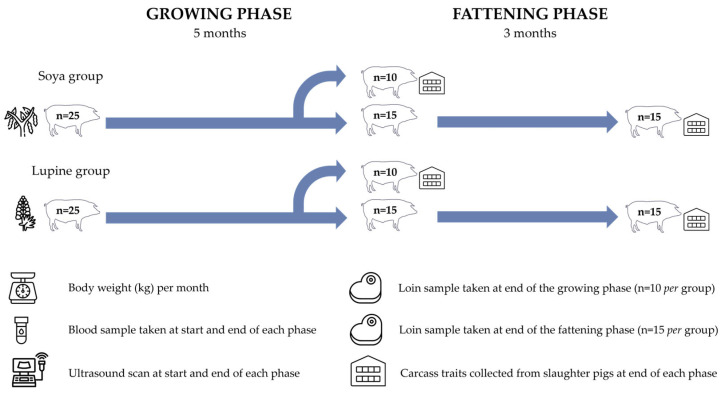
Experimental design.

**Table 1 animals-14-03084-t001:** Ingredients and nutrient composition of Iberian pig feeds.

	Pre-Finishing Diets		Fattening Diets	
	Soybean Diet	Lupine Diet	Soybean Diet	Lupine Diet
1. Ingredients (g/kg):				
Corn	319.3	319.3	269.2	254.2
Wheat	250	250	400	400
Barley	250	220	250	250
Soybean meal 47%	68.9	.	35.0	.
Lupine	.	105	.	50.0
Wheat Bran	75.4	69.3	.	.
Sugarcane molasses	10	10	.	.
Lard ^1^	.	.	20	20
Calcium carbonate	11.7	11.7	10.2	10.2
Sodium chloride	4.8	4.8	4.9	4.9
Dicalcium phosphate	4.4	4.4	5.9	5.9
L-lysine HCL	2.8	2.8	2.4	2.4
L-threonine	0.7	0.7	0.4	0.4
Vitamin-Mineral premix ^2^	2	2	2	2
2. Nutrients (g/kg):				
Dry matter	114.9	113.5	108.5	107.3
Ash	42.3	40.3	37.5	36.6
Crude protein	128.8	128.4	114.3	114.0
Crude fat	25.2	30.6	42.7	45.1
Crude fibre	35.7	45.9	30.8	36.1
Neutral detergent fibre	136.8	144.3	120.6	126.0
Net energy (kcal/kg)	1638.8	1648.1	1733.2	1736.4

^1^ Product of Intexur S.L. (Higuera la Real, Spain). It contains myristic acid (C14:0) 1.40%, palmitic acid (C16:0) 25.0%, palmitoleic acid (C16:1) 2.80%, stearic acid (C18:0) 12.8%, oleic acid (C18:1 n-9 cis) 47.2%, linoleic acid (C18:2 n-6 cis) 7.40%, α-linolenic acid (C18:3 n-3 cis) %, arachidonic acid (C20:4 n-6 cis) 0.10%, eicosapentaenoic acid (C20:5 n-3 cis) < 0.10%, docosahexaenoic acid (C22:6 n-3 cis) < 0.10%. ^2^ Product of Vilomix Iberia SLU (Barcelona, Spain) supplying per kilogram of feed: vitamin A (3a672a) 6500 UI; vitamin D3 (3a671) 1800 UI; vitamin E (3a700) 15 mg; vitamin K3 (3a710) 1.5 mg; vitamin B1 (3a821) 0.75 mg; vitamin B2 3 mg; vitamin B6 (3a831) 1 mg; vitamin B12 0.02 mg, biotin (3a880) 0.01 mg; niacin (3a314) 18 mg, calcium D-pantothenate (3a841) 10 mg; folic acid (3a316) 0.1 mg; choline chloride (3a890) 150 mg; Cu (3b405) (from CuSO_4_·5H_2_O) 15 mg; Zn (3b603) (from ZnO) 100 mg; Se (3b801) (from Na_2_SeO_3_) 0.3 mg; I (3b201) (from KI) 1 mg; Mn (3b502) (from MnO) 40 mg; Fe (3b103) (from FeSO_4_·H_2_O) 75 mg; butylhydroxytoluene (BHT) (E-321) 0.15 mg; propyl gallate (E-310) 0.02 mg; citric acid (E-330) 0.04 mg; sepiolite clay (E-563) 25 mg.

**Table 2 animals-14-03084-t002:** Effect of a lupine or a soya-based concentrate diet on the body weight (BW) of Iberian pigs during the growing and fattening phases.

Variable *	Soya Group	Lupine Group	RMSE	*p* Value
BW0	60.48	60.48	5.17	1
BW1	66.44	66.46	5.31	0.989
BW2	74.10	75.00	6.00	0.598
BW3	82.62	83.74	6.50	0.546
BW4	95.31	97.10	7.78	0.425
BW5	109.73	109.90	8.59	0.945
BW6	124.93	124.43	6.58	0.841
BW7	151.11	148.11	9.39	0.406
BW8	165.64	163.67	11.19	0.638

* BW during growth phase (BW0–BW5; n = 25/group) and fattening phase (BW5–BW8; n = 15/group).

**Table 3 animals-14-03084-t003:** Effect of a lupine or a soya-based concentrate diet on carcass traits predicted in vivo by ultrasound at the tenth rib level during the growing (n = 25/group) and fattening (n = 15/group) phases.

Variable	Soya Group	Lupine Group	RMSE *	*p* Value
Start of growing phase				
Loin depth (cm)	3.37	3.21	4.12	0.192
Loin area (cm^2^)	14.43	13.81	2.11	0.306
Back fat (cm)	1.79	1.82	3.01	0.742
End of growing phase				
Loin depth (cm)	4.50	4.68	4.82	0.194
Loin area (cm^2^)	19.46	18.82	2.33	0.348
Back fat (cm)	3.45	3.39	5.34	0.675
End of fattening phase				
Loin depth (cm)	6.02	6.21	5.73	0.387
Loin area (cm^2^)	25.66	26.38	2.94	0.590
Back fat (cm)	5.62	5.53	6.38	0.688

* RMSE: root mean square error.

**Table 4 animals-14-03084-t004:** Effect of a lupine or a soya-based concentrate diet on plasma biochemical parameters of Iberian pigs during the growing (n = 25/group) and fattening (n = 15/group) phases.

Variable	Soya Group	Lupine Group	RMSE *	*p* Value
Start of growing phase				
ALP (IU/L)	27.49	22.53	13.95	0.220
ALT (IU/L)	71.37	71.46	13.99	0.983
AST (IU/L)	117.23	128.91	91.08	0.656
GGT (IU/L)	181.00	158.64	109.48	0.483
Direct Bilirubin (mg/dL)	0.04	0.04	0.02	0.486
Cholesterol (mg/dL)	118.29	115.52	15.07	0.524
Triglycerides (mg/dL)	46.25	43.45	10.94	0.375
End of growing phase				
ALP (IU/L)	43.93	32.91	32.88	0.247
ALT (IU/L)	49.61	52.64	7.81	0.181
AST (IU/L)	56.69	67.48	21.50	0.089
GGT (IU/L)	112.12	81.04	56.72	0.061
Direct Bilirubin (mg/dL)	0.03	0.03	0.02	0.976
Cholesterol (mg/dL)	133.30	134.25	17.91	0.854
Triglycerides (mg/dL)	62.34	44.44	21.40	0.005
End of fattening phase				
ALP (IU/L)	33.10	34.60	24.50	0.876
ALT (IU/L)	71.62	60.46	6.61	0.001
AST (IU/L)	71.84	74.11	47.93	0.900
GGT (IU/L)	72.59	60.40	50.52	0.522
Direct Bilirubin (mg/dL)	0.03	0.03	0.02	0.241
Cholesterol (mg/dL)	186.74	185.59	37.92	0.935
Triglycerides (mg/dL)	74.47	78.10	38.11	0.800

* RMSE: root mean square error. ALP: alkaline phosphatase. ALT: alanine aminotransferase. AST: aspartate aminotransferase. GGT: gamma-glutamyl transferase.

**Table 5 animals-14-03084-t005:** Effect of a lupine or a soya-based concentrate diet group on carcass traits of Iberian *primor* (pre-finishing slaughtered) pigs (n = 10/group).

Variable	Soya Group	Lupine Group	RMSE *	*p* Value
Carcass weight (kg)	93.18	93.86	4.37	0.732
Carcass yield (%)	79.67	78.97	1.07	0.162
Prime cuts’ yield (%)	41.48	41.90	1.03	0.378
Ham weight (kg)	10.32	10.46	0.56	0.431
Ham yield (%)	22.15	22.31	0.71	0.626
Foreleg weight (kg)	7.38	7.58	0.38	0.105
Foreleg yield (%)	15.81	16.16	0.47	0.160
Loin weight (kg)	1.62	1.61	0.13	0.792
Loin yield (%)	3.48	3.43	0.25	0.662

* RMSE: root mean square error.

**Table 6 animals-14-03084-t006:** Effect of a lupine or a soya-based concentrate diet group on colour parameters, pH, intramuscular fat, and fatty acid content of the *Longissimus dorsi* muscle of Iberian *primor* (pre-finishing slaughtered) pigs (n = 10/group).

Variable	Soya Group	Lupine Group	RMSE *	*p* Value
L*	40.92	38.34	5.01	0.265
a*	9.18	9.96	2.33	0.462
b*	3.09	2.57	1.37	0.407
pH	5.86	5.90	0.28	0.750
IMF	4.64	3.77	0.47	0.058
SFA	37.91	35.86	1.45	0.007
MUFA	51.35	55.49	5.17	0.099
PUFA	10.74	8.64	5.21	0.394

* Colour: L* (lightness); a* (redness); b* (yellowness). IMF: intramuscular fat (%). SFA: saturated fatty acids (%). MUFA: monounsaturated fatty acids (%). PUFA: polyunsaturated fatty acids (%). RMSE: root mean square error.

**Table 7 animals-14-03084-t007:** Effect of a lupine or a soya-based concentrate diet on carcass traits of Iberian pigs after finishing (n = 15/group).

Variable	Soya Group	Lupine Group	RMSE *	*p* Value
Carcass weight (kg)	136.70	134.67	10.03	0.590
Carcass yield (%)	82.31	82.25	1.25	0.572
Prime cuts’ yield (%)	37.86	37.71	0.74	0.595
Ham weight (kg)	14.14	13.90	1.12	0.421
Ham yield (%)	20.67	20.65	0.50	0.901
Foreleg weight (kg)	9.92	9.72	0.71	0.293
Foreleg yield (%)	14.53	14.45	0.47	0.620
Loin weight (kg)	1.81	1.76	0.17	0.227
Loin yield (%)	2.65	2.62	0.22	0.654

* RMSE: root mean square error.

**Table 8 animals-14-03084-t008:** Effect of a lupine or a soya-based concentrate diet on colour parameters, pH, intramuscular fat, and fatty acid content of the *Longissimus dorsi* muscle of Iberian pigs after finishing (n = 15/group).

Variable	Soya Group	Lupine Group	RMSE *	*p* Value
L*	34.83	36.32	3.98	0.320
a*	7.21	7.32	1.46	0.847
b*	1.21	1.15	0.84	0.861
pH	5.88	6.02	0.22	0.103
IMF	8.22	9.26	2.02	0.180
SFA	37.00	38.15	4.65	0.513
MUFA	57.52	56.49	4.18	0.513
PUFA	5.48	5.36	1.08	0.776

* Colour: L* (lightness); a* (redness); b* (yellowness). IMF: intramuscular fat (%). SFA: saturated fatty acids (%). MUFA: monounsaturated fatty acids (%). PUFA: polyunsaturated fatty acids (%). RMSE: root mean square error.

## Data Availability

Dataset available on request from the authors.

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
