# Peer review of "Use of Lupinus albus as a Local Protein Source in the Production of High-Quality Iberian Pig Products"

_animals, 2024, doi:10.3390/ani14213084_

Round 1
Reviewer 1 Report
Comments and Suggestions for Authors
This study explores the potential of replacing soybean meal with sweet white lupin (Lupinus albus) in the diet of Iberian pigs, aiming to improve sustainability without compromising growth performance, carcass traits, or meat quality. The authors demonstrate that lupin-based diets yield comparable results to soybean meal diets while reducing the saturated fatty acid (SFA) content in meat, particularly in the pre-finishing phase, which may improve the health attributes of the meat.
Line 22: You should specify the sample size (n) for each group to clarify the number of pigs in each dietary treatment.
Line 36: You should specify the type of production system of Iberian pig.
Line 76: You should include citations of studies that address the presence of antinutritional factors in lupine and its limited use in Iberian pig production. Additionally, it would strengthen your argument to reference research on the development of new lupine varieties with lower levels of these factors and their potential inclusion in Iberian pig diets.
Line 125: The manuscript should clarify the meaning of "BW" (body weight) the first time it is mentioned to avoid any ambiguity for the reader.
Line 235: In Table 4, the manuscript should clearly specify what each unit of measurement refers to, making the table fully self-explanatory.
Abstract
The abstract effectively summarizes the study's objective, methods, and key findings. It clearly states that the study investigates the substitution of soybean meal with lupin in the diet of Iberian pigs, highlighting the main outcome of lower saturated fatty acid (SFA) levels in lupin-fed pigs. However, it lacks some details critical for a scientific abstract: you should include more specific information on sample size, statistical methods, and the significance (p-value) of the results.
Introduction
The introduction provides a good overview of the context, emphasizing the importance of sustainable pig production and the use of lupin as a local alternative to soybean meal. It also briefly introduces the relevance of Iberian pig production and consumer demands.
Materials and Methods
The methods section is detailed, describing the experimental design, diets, animal management, and measurements taken (e.g., body weight, carcass traits, meat quality, biochemical parameters). However, a few aspects could be enhanced:
Statistical Analysis: While ANOVA is mentioned, there could be more detail on how the assumptions of ANOVA were tested.
Results
The use of tables to present body weight, carcass traits, biochemical parameters, and fatty acid profiles is helpful for the reader. Results are organized logically, first discussing growth and carcass traits, followed by meat quality and biochemical parameters.
Discussion
Provide a more detailed exploration of the role and mechanisms of antinutritional factors in lupin, particularly focusing on how these factors have been reduced in the Lupinus albus variety used in the study. Although the manuscript briefly mentions that this variety is low in antinutritional factors, further elaboration is needed to explain the specific advancements in lupin breeding that have minimized these factors and the potential residual effects that may still persist.
Furthermore, the discussion would benefit from referencing additional studies to support the claim that antinutritional factors are no longer a concern. The manuscript currently mentions only a few studies, and one of them included lupin in a diet at a low rate (6%).
Conclusion
The conclusion is concise and effectively summarizes the key findings of the study, reiterating the potential of lupin to replace soybean meal in pig diets without negatively impacting production performance.
Author Response
REVIEWER 1
General comments
This study explores the potential of replacing soybean meal with sweet white lupin (Lupinus albus) in the diet of Iberian pigs, aiming to improve sustainability without compromising growth performance, carcass traits, or meat quality. The authors demonstrate that lupin-based diets yield comparable results to soybean meal diets while reducing the saturated fatty acid (SFA) content in meat, particularly in the pre-finishing phase, which may improve the health attributes of the meat.
AU: Dear anonymous Reviewer 1,
Thank you very much for your comments and for reviewing our manuscript in such detail! We have taken great care to acknowledge all your comments and consider your suggestions for improving the manuscript. Please find below our responses to each of the points raised.
Specific comments
Line 22: You should specify the sample size (n) for each group to clarify the number of pigs in each dietary treatment.
AU: Amended (new lines 21-22).
Line 36: You should specify the type of production system of Iberian pig.
AU: Amended (new line 38).
Line 76: You should include citations of studies that address the presence of antinutritional factors in lupine and its limited use in Iberian pig production. Additionally, it would strengthen your argument to reference research on the development of new lupine varieties with lower levels of these factors and their potential inclusion in Iberian pig diets.
AU: Included reference (new line 80).
Line 125: The manuscript should clarify the meaning of "BW" (body weight) the first time it is mentioned to avoid any ambiguity for the reader.
AU: Amended (new line 132).
Line 235: In Table 4, the manuscript should clearly specify what each unit of measurement refers to, making the table fully self-explanatory.
AU: Amended. The unit of measurement (UI/L) was incorrectly referenced in Spanish. It has been changed to international units per Liter (IU/L).
Abstract
The abstract effectively summarizes the study's objective, methods, and key findings. It clearly states that the study investigates the substitution of soybean meal with lupin in the diet of Iberian pigs, highlighting the main outcome of lower saturated fatty acid (SFA) levels in lupin-fed pigs. However, it lacks some details critical for a scientific abstract: you should include more specific information on sample size, statistical methods, and the significance (p-value) of the results.
AU: Amended (new lines 23-24).
Introduction
The introduction provides a good overview of the context, emphasizing the importance of sustainable pig production and the use of lupin as a local alternative to soybean meal. It also briefly introduces the relevance of Iberian pig production and consumer demands.
Materials and Methods
The methods section is detailed, describing the experimental design, diets, animal management, and measurements taken (e.g., body weight, carcass traits, meat quality, biochemical parameters). However, a few aspects could be enhanced:
Statistical Analysis: While ANOVA is mentioned, there could be more detail on how the assumptions of ANOVA were tested.
Results
The use of tables to present body weight, carcass traits, biochemical parameters, and fatty acid profiles is helpful for the reader. Results are organized logically, first discussing growth and carcass traits, followed by meat quality and biochemical parameters.
Discussion
Provide a more detailed exploration of the role and mechanisms of antinutritional factors in lupin, particularly focusing on how these factors have been reduced in the Lupinus albus variety used in the study. Although the manuscript briefly mentions that this variety is low in antinutritional factors, further elaboration is needed to explain the specific advancements in lupin breeding that have minimized these factors and the potential residual effects that may still persist. Furthermore, the discussion would benefit from referencing additional studies to support the claim that antinutritional factors are no longer a concern. The manuscript currently mentions only a few studies, and one of them included lupin in a diet at a low rate (6%).
AU: Amended (new lines 93-95).
Conclusion
The conclusion is concise and effectively summarizes the key findings of the study, reiterating the potential of lupin to replace soybean meal in pig diets without negatively impacting production performance.
Reviewer 2 Report
Comments and Suggestions for Authors
Lupinus albus, a local protein source that can replace soybean meal in pig production, was assessed for this purpose in the research. It is important for investigating sustainable feed sources and improving meat quality in Iberian production. However, the experimental design and statistics have significant flaws because each group has 25 or 15 pigs during the growth and fattening stages, respectively. Consequently, rather than the individuals, each group only has one observation in the average. Since all of the samples or pigs are a part of a single replication, statistical analyses are not feasible, and it is challenging to discuss the findings and conclusions without a solid scientific foundation.
Author Response
REVIEWER 2
General comments
Lupinus albus, a local protein source that can replace soybean meal in pig production, was assessed for this purpose in the research. It is important for investigating sustainable feed sources and improving meat quality in Iberian production. However, the experimental design and statistics have significant flaws because each group has 25 or 15 pigs during the growth and fattening stages, respectively. Consequently, rather than the individuals, each group only has one observation in the average. Since all of the samples or pigs are a part of a single replication, statistical analyses are not feasible, and it is challenging to discuss the findings and conclusions without a solid scientific foundation.
AU: Dear anonymous Reviewer 2,
Thank you for taking the time to read this and for your comment. In the experiment, a total of 50 Iberian pigs were randomly assigned to two dietary groups (n=25 per group). The experimental design included two production phases: growth and fattening. During the growth phase, a subset of animals from each group was slaughtered in order to analyse the meat characteristics at the end of this phase. The remaining animals continued through the fattening phase, where meat analysis was again performed at the end of the study.
Thus, we have two observation points within each group, corresponding to the animals slaughtered at the end of the growth phase and those slaughtered at the end of the fattening phase. Although the number of animals in each subpopulation was reduced at each stage, the design was planned to capture the evolution of the animals' characteristics throughout these phases.
In terms of statistical analysis, we acknowledge the reduced sample size in the subpopulations. However, we believe that the data collected from both phases are representative of the performance of each group and have been appropriately analysed using the statistical methods described. We believe that the results provide relevant information on the effects of the diets at different phases of production.
Reviewer 3 Report
Comments and Suggestions for Authors
This study investigated the effects of Lupinus albus on the growth and metabolic parameters, carcass yield and the meat quality of Iberian pigs. The research results offer the possibility of Lupinus albus replacing soybean protein sources and provide a reference for the local production of Iberian pigs. The experimental design of this study is reasonable. The research data are comprehensive and reliable, and the writing is relatively standard. I have only two small modification suggestions as follows:
Line 165-169: “The yields of carcass, hams, forelegs, loins, and total prime cuts were calculated as done by [35]” The definitions of these cuts (the specific positions where they are made) need to be elaborated in detail because this is part of the methodology. The literature cited by the author is a conference paper, and the reviewer cannot find the original text of the conference paper.
Line 209: “with a significance level of P < 0.05.” A P value less than 0.05 is certainly fine as the significance criterion. However, due to the small sample size of this study, results with a P value less than 0.1 can also be considered as having a trend of significant difference. Especially for some results with P values close to 0.05, more consideration should be given. For example, in Table 4, the P value of Triglycerides is 0.05, but the P value of GGT is 0.061, which is very close to 0.05; in Table 6, the P value of IMF is 0.058. If the author considers that results with a P value less than 0.1 can also be regarded as having a trend of significant difference in the statistics, then relevant indicators can be mentioned in the result description and the subsequent discussion and conclusions. This would make the research results more comprehensive.
Author Response
REVIEWER 3
General comments
This study investigated the effects of Lupinus albus on the growth and metabolic parameters, carcass yield and the meat quality of Iberian pigs. The research results offer the possibility of Lupinus albus replacing soybean protein sources and provide a reference for the local production of Iberian pigs. The experimental design of this study is reasonable. The research data are comprehensive and reliable, and the writing is relatively standard. I have only two small modification suggestions as follows:
AU: Dear anonymous Reviewer 3,
Thank you for taking the time to read this and for your valuable comments. We hope that this revised version meets the reviewer's expectations.
Specific comments
Line 165-169: “The yields of carcass, hams, forelegs, loins, and total prime cuts were calculated as done by [35]” The definitions of these cuts (the specific positions where they are made) need to be elaborated in detail because this is part of the methodology. The literature cited by the author is a conference paper, and the reviewer cannot find the original text of the conference paper.
AU: Amended (new lines 174-188).
Line 209: “with a significance level of P < 0.05.” A P value less than 0.05 is certainly fine as the significance criterion. However, due to the small sample size of this study, results with a P value less than 0.1 can also be considered as having a trend of significant difference. Especially for some results with P values close to 0.05, more consideration should be given. For example, in Table 4, the P value of Triglycerides is 0.05, but the P value of GGT is 0.061, which is very close to 0.05; in Table 6, the P value of IMF is 0.058. If the author considers that results with a P value less than 0.1 can also be regarded as having a trend of significant difference in the statistics, then relevant indicators can be mentioned in the result description and the subsequent discussion and conclusions. This would make the research results more comprehensive.
AU: Thank you for your comment. However, I would like to clarify that, as shown in Table 4, the P value for triglycerides is 0.005, not 0.05. Regarding the GGT values, I did not highlight a trend towards significance because both study groups were within the standard reference ranges for this parameter. In addition, GGT levels were slightly higher in the soybean group than in the lupin group. Despite this difference, the values for both groups remained within normal limits, so I did not highlight this in the discussion.
Reviewer 4 Report
Comments and Suggestions for Authors
Dear Authors,
The manuscript is is a good work with clear objectives and which provides an advance in existing knowledge.
The results are clearly presented.Best regards
Author Response
REVIEWER 4
General comments
Dear Authors,
The manuscript is a good work with clear objectives and which provides an advance in existing knowledge.
The results are clearly presented.
Best regards
AU: Dear anonymous Reviewer 4,
Thank you very much for your comments and for reviewing our manuscript.
Round 2
Reviewer 2 Report
Comments and Suggestions for Authors
Different from the characteristics of meat quality and serum parameters, it is impracticable to compare the growth performance statistically using only one or two observations of every treatment, statistical analysis necessitates at least three replicates of each experimental group.
Therefore, methods should include information on the number of pans in total to use, and how many pigs of each pan used, and how to monitor feed intake and body weight.
Author Response
AU: Dear anonymous Reviewer 2,
Thank you for your continued review and valuable feedback. We understand your concerns regarding the statistical analysis of growth performance with the current experimental design. Below we provide further clarification on how the growth performance data were collected and analysed.
In our study, pigs were divided into two dietary groups (n=25 per group), with each group housed in a separate pen. Due to the nature of the study, where animals were monitored under semi-extensive conditions, individual pigs were not housed in separate pens, which limited the ability to make individual measurements of feed intake per pig. Instead, feed intake was monitored at the group level, while body weight was measured individually at regular intervals throughout the experiment.
The design was constrained by the housing conditions and the need to evaluate the pigs under more natural semi-extensive farming conditions, which better reflect commercial Iberian pig production. For the analysis of growth performance, body weight was measured monthly, which allowed us to follow the evolution of weight gain over the production phases (growing and fattening). However, we agree that the lack of individual monitoring of feed intake introduces some limitations.
We would also like to emphasise that we believe that the information provided by the current design, particularly about meat quality, ultrasound measurements and biochemical parameters, remains valuable for understanding the effects of the dietary treatments.
If you feel that further clarification or adjustment is needed in the methods section, we would be happy to revise the manuscript accordingly to better explain the limitations of the growth performance analysis and how feed intake and body weight were monitored.